# Non-Covalent Interactions in Triglycerides: Vaporisation Thermodynamics for Quantification of Dispersion Forces

Sergey P. Verevkin [1,2,*] and Ruslan N. Nagrimanov [2]

1. Department of Physical Chemistry and Faculty of Interdisciplinary Research, Competence Centre CALOR, University of Rostock, 18059 Rostock, Germany
2. Department of Physical Chemistry, Kazan Federal University, 420008 Kazan, Russia
* Correspondence: sergey.verevkin@uni-rostock.de

**Abstract:** Qualitatively, the non-covalent interactions are well-known and help to explain many phenomena in chemistry and biochemistry. Quantitatively, determination of strength this force is a challenging task. The vaporization enthalpy is a reliable measure not only for the intermolecular interactions in the liquid phase, but also as the measure of intermolecular non-covalent interactions in the gas phase for the specific group of compounds, e.g., for the triglycerides. The vaporisation thermodynamics of four triglycerides were studied by using transpiration method, quartz crystal microbalance, and thermogravimetric analysis. Vapour pressure–temperature dependences were used to derive the enthalpies of vaporisation of these very low volatile liquids. Vaporisation enthalpies of the triglycerides available in the literature were collected and uniformly adjusted to the reference temperature 298.15 K and validated using structure–property relationships (chain-length dependence, correlation with retention indices, and correlation with normal boiling points). The consistent sets of evaluated vaporisation enthalpies for the linear and branched triglycerides were used to develop the "centerpiece" based group-additivity method for predicting enthalpies of vaporisation of triglycerides. It has turned out that the family of triglycerides do not obey the group-additivity rules. The reason for that is that the evaporated in the gas phase triglycerides exhibit intensive non-covalent attractive dispersion interactions strongly dependent on the alkyl-chain length. For the first time the intensity of the dispersion interactions was quantified for the family of aliphatic linear triglycerides with the chain length from 3 to 18 carbon atoms. The influence of the branching and unsaturation of the alkyl chains to the strength of the non-covalent interactions was also discussed.

**Keywords:** non-covalent interaction; vapour pressure measurements; enthalpy of vaporisation; structure–property relationships; group-additivity

## 1. Introduction

Although non-covalent interactions play a key role in material science, chemistry and biochemistry, their interpretation and quantification are still far from being satisfactory. Dispersion forces are much more difficult to handle and therefore less is known about them, in particularly quantitatively. The dispersion forces are usually related to the attractive part of the van der Waals potential [1]. The simplest example to show the importance of dispersion forces is that they help explain why alkanes become liquid with increasing chain length. Admittedly, the dispersion interactions are considered "weak". Because of this, quantifying dispersion forces is quite challenging. However, the dispersion interactions increase rapidly for larger and larger molecules [2]. In our recent work, very weak dispersion forces were quantified in methyl alkanoates with an alkyl chain length of 1 to 18 carbon atoms [3]. The homologous series of triglycerides offers a three-fold increase in size compared to the ester series. We expect a significant increase in the role of dispersion forces in triglycerides and are looking for thermodynamic tools to quantify these non-covalent interactions.

In this article, we have carefully collected experimental data on vaporisation thermodynamics for triglycerides that are available in the literature. To ascertain the vapour

pressures and vaporisation enthalpies, complementary vapour pressure measurements were carried out on a number of triglycerides (see Figure 1).

**Figure 1.** General structures (**left**) and abbreviation (**right**) of triglycerides evaluated in this work. The trivial names of the triglycerides such as triacetin, tricaprylin, tricaprin, and tripalmitin are somewhat awkward for systematic data analysis. In our opinion, the numerical description of the chain lengths attached to the glycerol moiety is more convenient. To keep consistency with our previous work [4], the designation of triglycerides has been ascribed as follows triacetin (TG $2_0 2_0 2_0$), tricaprylin (TG $8_0 8_0 8_0$), tricaprin (TG $10_0 10_0 10_0$), and tripalmitin (TG $16_0 16_0 16_0$). Such a designation is particularly useful when the triglyceride has double bonds in the alkyl chains (see Figure S1). To help readers, the IUPAC and the trivial nomenclatures are given for each triglyceride in the supplementary materials.

We have modified the Chickos's method [5] for calculations of $\Delta_l^g C_{p,m}^o$-values, required for the temperature adjustments of experimental vaporisation enthalpies. The available vaporisation enthalpies of triglycerides were collected, uniformly adjusted to the reference temperature $T = 298.15$ K and evaluated using the structure–property correlations based on the vaporisation enthalpy chain length dependency, retention indices, boiling temperatures, and group additivity. The reliable data sets for the triglycerides with the linear and branched chains have been recommended for thermochemical calculations. The recommended experimental data in combination with a group-additivity based "centerpiece" approach were used to reveal and quantify dispersion forces in triglycerides.

## 2. Materials and Methods

The samples of triglycerides triacetin, tricaprylin, tricaprin, and tripalmitin of commercial origin (Sigma-Aldrich, 99%) were used as received. The degree of purity was determined using a Hewlett Packard gas chromatograph 5890 Series II equipped with a flame ionization detector. A capillary column SE-30 was used with a column length of 10 m, an inside diameter of 0.32 mm, and a film thickness of 0.25 μm. The standard temperature program of the GC was $T = 373$ K followed by a heating rate of 0.167 K·s$^{-1}$ to $T = 523$ K. No impurities (greater than mass fraction 0.001) could be detected in the sample used for the vapour pressure measurements. Before starting the vapour pressure measurements, the sample was preconditioned inside of the set-up to remove traces of water and possible volatile impurities.

The standard molar enthalpies of vaporisation, $\Delta_l^g H_m^o$, of triglycerides were derived from the temperature dependences of vapour pressures measured using the transpiration method [6,7] and the quartz-crystal microbalance (QCM) method [8]. The temperature dependences of the mass loss rates measured using termogravimetric analysis (TGA) [9] were used to derive $\Delta_l^g H_m^o$-values of triacetin, tricaprylin, and tricaprin. A concise description of the experimental methods and data treatment is given in the Supporting Information.

### 3. Results and Discussion

#### 3.1. Experimental Vaporisation Thermodynamics of Triglycerides

The original experimental vapour pressures of triglycerides at different temperatures measured using transpiration method are collected in Table S1. The original experimental vapour pressures of triglycerides at different temperatures measured using QCM method are collected in Table S2. The mass-loss rates of triglycerides at different temperatures measured using TGA method are collected in Table S3. These results were used to derive the standard molar enthalpies of vaporisation $\Delta_l^g H_m^o(T_{av})$ which are referenced to the average temperatures $T_{av}$. These results are shown in Table 1, column 4. For thermochemical calculations, the vaporisation enthalpies are used to adjust to the reference temperature $T$ = 298.15 K according to the Kirchhoff's equation:

$$\Delta_l^g H_m^o(298.15 \text{ K}) = \Delta_l^g H_m^o(T_{av}) + \Delta C_{p,m}^o \times (T_{av} - 298.15 \text{ K}) \tag{1}$$

where the value $\Delta_l^g C_{p,m}^o = C_{p,m}^o(g) - C_{p,m}^o(liq)$ is the difference between the molar heat capacities of the gaseous $C_{p,m}^o(g)$ and the liquid phase $C_{p,m}^o(liq)$, respectively. The required $\Delta_l^g C_{p,m}^o$-values are evaluated in Section 3.2.

**Table 1.** Compilation of available enthalpies of vaporisation $\Delta_l^g H_m^o$ of linear triglycerides.

| Compound CAS | Method [a] | T-Range K | $\Delta_l^g H_m^o(T_{av})$ kJ·mol$^{-1}$ | $\Delta_l^g H_m^o$(298.15 K) kJ·mol$^{-1}$ | Ref. |
|---|---|---|---|---|---|
| TG $2_0 2_0 2_0$ | S | 284.2–318.2 | 82.0 ± 0.5 | 82.3 ± 0.6 | [10] |
| 102-76-1 | C | 298.15 | | 83.4 ± 2.0 | [11] |
| triacetin | C | 298.15 | | (85.7 ± 0.6) | [12] |
| | E | 439.5–590.2 | 59.7 ± 0.9 | 78.7 ± 3.9 | [13] |
| | T | 320.1–360.9 | 77.1 ± 0.4 | 80.8 ± 0.5 | [14] |
| | T | 300.2–328.2 | 82.3 ± 0.8 | 83.6 ± 0.9 | [15] |
| | T | 318.1–362.9 | 76.4 ± 0.5 | 80.2 ± 0.9 | Table S1 |
| | | | | **81.5 ± 0.3** [b] | average |
| | $n_c$ | | | 79.3 ± 3.0 | Table 4 |
| | $J_x$ | | | 80.3 ± 3.0 | Table 5 |
| | $T_b$ | | | 79.4 ± 3.0 | Table 6 |
| TG $3_0 3_0 3_0$ | T | 304.2–337.2 | 88.1 ± 0.4 | 90.3 ± 0.5 | [15] |
| 139-45-7 | BP | 403–545 | 77.1 ± 1.8 | 94.7 ± 3.9 | Table S4 |
| tripropionin | | | | **90.4 ± 0.5** [b] | average |
| | $n_c$ | | | 88.9 ± 3.0 | Table 4 |
| | $J_x$ | | | 89.5 ± 3.0 | Table 5 |
| | $T_b$ | | | 90.2 ± 3.0 | Table 6 |
| TG $4_0 4_0 4_0$ | C | 298.15 | | 107.1 ± 5.0 | [12] |
| 60-01-5 | S | 318–364 | 81.2 ± 3.0 | (86.2 ± 3.2) | [16,17] |
| tributyrin | ITGA | 349.3 | 99.9 ± 3.5 | 105.6 ± 3.7 | [18] |
| | TGA | 323–593 | 78.4 ± 5.0 | 94.1 ± 5.9 | [19] |
| | NTGA | 476.8–584.4 | 78.4 ± 0.7 | 105.5 ± 5.5 | [20] |
| | NTGA | 308.6 | 83.5 ± 0.4 | (84.8 ± 2.5) | [21] |
| | ITGA | 373 | 84.9 ± 3.5 | 93.8 ± 3.9 | [22] |
| | T | 324.2–354.2 | 92.2 ± 0.5 | 97.1 ± 0.6 | [15] |
| | BP | 394–583 | 81.4 ± 0.7 | 102.7 ± 4.3 | Table S4 |
| | | | | **97.5 ± 0.6** [b] | average |
| | $n_c$ | | | 98.4 ± 3.0 | Table 4 |
| | $J_x$ | | | 98.1 ± 3.0 | Table 5 |
| | $T_b$ | | | 97.4 ± 3.0 | Table 6 |

**Table 1.** *Cont.*

| Compound | Method [a] | $T$-Range | $\Delta_l^g H_m^o(T_{av})$ | $\Delta_l^g H_m^o$(298.15 K) | Ref. |
|---|---|---|---|---|---|
| CAS | | K | kJ·mol$^{-1}$ | kJ·mol$^{-1}$ | |
| TG $5_0 5_0 5_0$ | T | 340.0–370.0 | 97.5 ± 0.5 | **104.8 ± 0.6** | [15] |
| 620-68-8 | $n_c$ | | | 107.9 ± 3.0 | Table 4 |
| tripentanoin | $J_x$ | | | 107.5 ± 3.0 | Table 5 |
| TG $6_0 6_0 6_0$ | S | 359–410 | 94.0 ± 3.0 | (106.0 ± 3.8) | [16,17] |
| 621-70-5 | ITGA | 386.1 | 118.7 ± 4.2 | (131.1 ± 4.9) | [18] |
| tricapronin | TGA | 353–653 | 92.8 ± 5.0 | 117.4 ± 7.0 | [19] |
| | NTGA | 511.0–641.3 | 78.0 ± 2.6 | 116.8 ± 8.2 | [23] |
| | NTGA | 349.4 | 99.9 ± 2.2 | (107.1 ± 2.6) | [21] |
| | NTGA | 519.2–646.4 | 70.9 ± 5.0 | 110.4 ± 9.3 | [24] |
| | ITGA | 373 | 96.4 ± 4.2 | (107.0 ± 4.7) | [22] |
| | BP | 428–633 | 84.2 ± 3.3 | 114.2 ± 5.6 | Table S4 |
| | | | | **114.9 ± 3.6** [b] | average |
| | $n_c$ | | | 117.5 ± 3.0 | Table 4 |
| | $J_x$ | | | 116.9 ± 3.0 | Table 5 |
| | $T_b$ | | | 115.3 ± 3.0 | Table 6 |
| TG $7_0 7_0 7_0$ | S | 401.7–452.2 | 80.8 ± 1.6 | (100.9 ± 4.3) | [25] |
| 620-67-7 | $n_c$ | | | 127.0 ± 3.0 | Table 4 |
| triheptanoin | $J_x$ | | | 127.2 ± 3.0 | Table 5 |
| | $T_b$ | | | 126.8 ± 3.0 | Table 6 |
| | | | | **127.0 ± 1.7** [b] | average |
| TG $8_0 8_0 8_0$ | S | 396–453 | 115.8 ± 3.0 | 138.4 ± 5.4 | [16,17] |
| 538-23-8 | ITGA | 411.4 | 130.0 ± 4.6 | 150.3 ± 6.1 | [18] |
| tricaprylin | TGA | 398–623 | 117.2 ± 5.3 | 152.6 ± 8.8 | [19] |
| | NTGA | 551.6–658.5 | 103.8 ± 9.2 | 158 ± 14 | [23] |
| | NTGA | 386.2 | 118.7 ± 4.7 | 134.5 ± 5.7 | [21] |
| | NTGA | 562.2–657.8 | 104.0 ± 5.0 | 159 ± 12 | [24] |
| | ITGA | 373 | 119.4 ± 4.6 | 132.8 ± 5.3 | [22] |
| | ITGA | 398.3–461.6 | 111.7 ± 0.5 | 135.0 ± 4.7 | Table S3 |
| | T | 403.2–448.2 | 109.2 ± 0.5 | 131.9 ± 1.4 | Table S1 |
| | | | | **134.1 ± 1.2** [b] | average |
| | $n_c$ | | | 136.6 ± 3.0 | Table 4 |
| | $J_x$ | | | 137.4 ± 3.0 | Table 5 |
| | $T_b$ | | | 136.1 ± 3.0 | Table 6 |
| TG $10_0 10_0 10_0$ | S | 437–485 | 124.5 ± 3.0 | 157.9 ± 7.3 | [16,17] |
| 621-71-6 | ITGA | 437.9 | 147.1 ± 5.1 | (175.7 ± 7.7) | [18] |
| tridecanoin | TGA | 443–673 | 138.6 ± 5.0 | (189 ± 11) | [19] |
| | NTGA | 597.6–692.5 | 130.3 ± 1.5 | (201 ± 14) | [20] |
| | NTGA | 411.5 | 130.5 ± 7.0 | 153.8 ± 8.4 | [21] |
| | I-TGA | 427.9–491.6 | 136.2 ± 0.5 | 169.6 ± 4.7 | Table S3 |
| | QCM | 348.5–367.8 | 154.5 ± 0.9 | 166.7 ± 2.6 | Table S2 |
| | T | 418.2–468.2 | 129.5 ± 0.8 | 159.0 ± 1.0 | Table S1 |
| | | | | **160.1 ± 0.9** [b] | average |
| | $n_c$ | | | 155.6 ± 3.0 | Table 4 |
| | $J_x$ | | | 156.1 ± 3.0 | Table 5 |
| TG $12_0 12_0 12_0$ | S | 458–520 | 137.4 ± 3.0 | (187 ± 10) | [16,17] |
| 538-24-9 | ITGA | 468.7 | 155.8 ± 5.5 | (200 ± 10) | [18] |
| trilaurin | NTGA | 438.0 | 147 ± 11 | (183 ± 13) | [21] |
| | NTGA | 615–667 | 221 ± 10 | (310 ± 20) | [26] |
| | $n_c$ | | | 174.7 ± 3.0 | Table 4 |
| | $J_x$ | | | 174.9 ± 3.0 | Table 5 |
| | | | | **174.9 ± 2.1** [b] | average |

**Table 1.** *Cont.*

| Compound CAS | Method [a] | T-Range K | $\Delta_l^g H_m^o(T_{av})$ kJ·mol$^{-1}$ | $\Delta_l^g H_m^o$(298.15 K) kJ·mol$^{-1}$ | Ref. |
|---|---|---|---|---|---|
| $TG\ 14_014_014_0$ | S | 458–520 | 137.0 ± 3.0 | 196.4 ± 12.1 | [16,17] |
| 555-45-3 | ITGA | 483.1 | 166.3 ± 5.8 | (224 ± 13) | [18] |
| trimyristin | NTGA | 468.9 | 155.8 ± 15.9 | (209 ± 19) | [21] |
| | NTGA | 615–660 | 198 ± 10 | (304 ± 23) | [26] |
| | $n_c$ | | | 193.8 ± 3.0 | Table 4 |
| | $J_x$ | | | 194.0 ± 3.0 | Table 5 |
| | | | | **194.0 ± 2.1** [b] | average |
| $TG\ 16_016_016_0$ | S | 506–572 | 160.6 ± 3.0 | (249 ± 18) | [16,17] |
| 555-44-2 | ITGA | 505.8 | 174.9 ± 6.1 | (252 ± 17) | [18] |
| tripalmitin | NTGA | 483.2 | 166 ± 20 | (235 ± 25) | [21] |
| | NTGA | 615–667 | 474 ± 19 | (601 ± 27) | [26] |
| | QCM | 384.9–433.2 | 169.5 ± 4.2 | 210.2 ± 9.2 | Table S2 |
| | $n_c$ | | | 212.9 ± 3.0 | Table 4 |
| | $J_x$ | | | 213.0 ± 3.0 | Table 5 |
| | $T_b$ | | | 209.0 ± 3.0 | Table 6 |
| | | | | **211.6 ± 1.7** [b] | average |
| $TG\ 18_018_018_0$ | S | 521–588 | 167.4 ± 3.0 | (182.9 ± 4.4) | [16,17] |
| 555-43-1 | NTGA | 505.9 | 175 ± 23 | (267 ± 30) | [18] |
| tristearin | NTGA | 610–660 | 221 ± 10 | (367 ± 31) | [26] |
| | $n_c$ | | | 232.0 ± 3.0 | Table 4 |
| | $J_x$ | | | 232.1 ± 3.0 | Table 5 |
| | | | | **232.1 ± 2.1** [b] | average |

[a] Methods: T = transpiration; S = static method; C = calorimetry; $J_x$—from correlation of experimental vaporisation enthalpies with Kovats indices (see text); BP—from experimental boiling temperatures reported at different pressures compiled from the literature (see Table S4); $T_b$ = from correlation of vaporisation enthalpies with the normal boiling points; E = ebulliometry; ITGA = isothermal TGA; NTGA = non-isothermal TGA; QCM = quartz-crystal microbalance. Uncertainties in the temperature adjustment of vaporisation enthalpies, are estimates and amount to 20% of the total adjustment. [b] Weighted mean value (the uncertainty was taken as the weighing factor). Uncertainty of the vaporisation enthalpy is expressed as the expanded uncertainty (0.95 level of confidence, k = 2). Values in parentheses were not considered. Values highlighted in bold were recommended for thermochemical calculations.

In this study, we carefully collected and evaluated the available experimental literature data on vapour pressures of triglycerides with linear alkyl chains (see Table 1) and with branched alkyl chains (see Table 2). Since in most studies the enthalpies of vaporisation were not adjusted to the reference temperature or the adjustment was performed in some other way, we treated the literature results with Equation (1) and calculated $\Delta_l^g H_m^o$-values for comparison and evaluation (see Tables 1 and 2, column 5).

**Table 2.** Compilation of available enthalpies of vaporisation $\Delta_l^g H_m^o$ of branched triglycerides.

| Compound CAS | M [a] | T- Range K | $\Delta_l^g H_m^o(T_{av})$ kJ·mol$^{-1}$ | $\Delta_l^g H_m^o$(298.15 K) kJ·mol$^{-1}$ | Ref. |
|---|---|---|---|---|---|
| glycerol triformate | T | 307.2–333.2 | 76.6 ± 1.1 | **78.2 ± 1.2** | [15] |
| 32765-69-8 | $J_x$ | | | 73.9 ± 3.0 | Table 5 |
| | GA | | | 74.5 [b] | this work |
| glycerol tri(2-methylpropanoate) | T | 329.1–371.2 | 87.8 ± 0.6 | **93.6 ± 0.7** | [15] |
| 14295-64-8 | $J_x$ | | | 93.8 ± 3.0 | Table 5 |
| | GA | | | 96.0 [b] | this work |
| glycerol tri(3-methylbutanoate) | T | 341.0–369.0 | 96.2 ± 1.2 | **104.0 ± 1.3** | [15] |
| 620-63-3 | $J_x$ | | | 102.8 ± 3.0 | Table 5 |
| | $T_b$ | | | 105.5 ± 3.0 | Table 6 |
| | GA | | | 104.1 [b] | this work |

**Table 2.** *Cont.*

| Compound | M [a] | *T*- Range | $\Delta_l^g H_m^o(T_{av})$ | $\Delta_l^g H_m^o(298.15\ K)$ | Ref. |
|---|---|---|---|---|---|
| CAS | | K | kJ·mol$^{-1}$ | kJ·mol$^{-1}$ | |
| glycerol tri(2,2-dimethylpropanoate) | T | 313.4–358.1 | 85.3 ± 0.8 | **90.2 ± 0.9** | [15] |
| 58006-18-1 | $J_x$ | | | 96.1 ± 3.0 | Table 5 |
| | GA | | | 97.0 [b] | this work |
| glycerol tribenzoate | S | 423–476 | 123.5 ± 3.0 | **141.4 ± 4.7** | [16,17] |
| 614-33-5 | GA | | | 150.7 [b] | this work |
| glycerol trierucate (TG 22$_1$,22$_1$,22$_1$) | QCM | 433.7–483.7 | 215.0 ± 4.0 | **310 ± 19** | [2] |
| 2752-99-0 | GA | | | 354.4 [b] | this work |

[a] Methods: T = transpiration; S = static method; $J_x$—from correlation of experimental vaporisation enthalpies with Kovats's indices (see text); $T_b$ = from correlation of vaporisation enthalpies with the normal boiling points; GA = estimated using group-additivity (see text); QCM = quartz-crystal microbalance. Uncertainties in the temperature adjustment of vaporisation enthalpies, are estimates and amount to 20% of the total adjustment.
[b] Calculated using increments listed in Table 7.

*3.2. Adjustment of $\Delta_l^g H_m^o(T)$-Values to the Reference Temperature 298.15 K*

In general, the adjustment of the thermodynamic properties to the reference temperature *T* = 298.15 K is important for the comparison and the development of the structure–property relationships. Admittedly [3,27,28], the vaporisation enthalpies have mostly been reported by authors as referenced to the $T_{av}$, and they have not often been adjusted to a different temperature apparently, due to the ambiguities with the $\Delta_l^g C_{p,m}^o$—values required in Equation (1). This ambiguity was resolved in systematic studies by Chickos and Acree [5,29] who proposed estimating heat capacity differences using the following empirical correlation:

$$-\Delta_l^g C_{p,m}^o = 10.58 + 0.26 \times C_{p,m}^o(\text{liq}) \tag{2}$$

which has been parameterized in general with the available data on the organic compounds of different classes. From our experience, the parameters of Equation (2) apply successfully to many classes of organic compounds successfully [3,27,28]. However, in our recent study on linear aliphatic esters (where the $\Delta_l^g C_{p,m}^o$-values were derived from temperature dependences vapour pressures, see Table S5), we have found that the original coefficients of Equation (2) provide significantly overestimated $\Delta_l^g C_{p,m}^o$-values [3]. In this work we correlated experimental the $C_{p,m}^o(\text{liq})$ and the $\Delta_l^g C_{p,m}^o$-values for linear aliphatic esters (see Table S5) and obtained the following empirical equation:

$$-\Delta_l^g C_{p,m}^o = 16.4 + 0.1833 \times C_{p,m}^o(\text{liq})\ (\text{with } R^2 = 0.979) \tag{3}$$

Both empirical coefficients are significantly lower than the original values from Chickos and Acree [5,29], but the high correlation coefficient $R^2$ is evidence for the robustness of the correlation according to Equation (3). Perhaps, the reason for the deviation of the empirical coefficients from those of the original values is that not too many long-chain species were included in the evaluation of Chickos and Acree [5,29]. It seems that for molecules with the monotonically growing alkyl chain, there are some peculiarities that should be taken into account. This observation should be validated with classes of organic compounds other than esters. However, since the triglycerides are most closely related to the long-chain esters, we decided to apply Equation (3) to estimate the $\Delta_l^g C_{p,m}^o$-values for this class as well.

Now, the molar heat capacities $C_{p,m}^o(\text{liq})$ of triglycerides are required to apply Equation (3) and calculate the desired $\Delta_l^g C_{p,m}^o$-values for the temperature adjustment of vaporisation enthalpies. The compilation of the $C_{p,m}^o(\text{liq})$-values available in the literature is given in Table 3.

**Table 3.** Compilation of data on molar heat capacities $C_{p,m}^o$ (liq) and heat capacity differences $\Delta_l^g C_{p,m}^o$ for triglycerides at $T = 298.15$ K (in J.K$^{-1}$.mol$^{-1}$).

| Compounds | $N_C$ [a] | $C_{p,m}^o$(liq) | $-\Delta_l^g C_{p,m}^o$ [c] |
|---|---|---|---|
| glycerol triformate | 1 | 284 [b] | 69 |
| TG $2_0 2_0 2_0$ | 2 | **389.0 [12]** | 88 |
| TG $3_0 3_0 3_0$ | 3 | **481.3 [12]** | 105 |
| TG $4_0 4_0 4_0$ | 4 | **555.3 [12]** | 118 |
| glycerol tri(2-methylpropanoate) | 4 | 561 [b] | 119 |
| TG $5_0 5_0 5_0$ | 5 | 610 [b] | 128 |
| glycerol tri(3-methylbutanoate) | 5 | 657 [b] | 137 |
| glycerol tri(2,2-dimethylpropanoate) | 5 | 640 [b] | 134 |
| TG $6_0 6_0 6_0$ | 6 | 682 [b] | 141 |
| tribenzoin | 7 | 555 [b] | 118 |
| TG $7_0 7_0 7_0$ | 7 | 765 [b] | 157 |
| TG $8_0 8_0 8_0$ | 8 | **886 [30]** | 179 |
| TG $10_0 10_0 10_0$ | 10 | **1028 [30]** | 205 |
| TG $12_0 12_0 12_0$ | 12 | **1322 [30]** | 259 |
| TG $14_0 14_0 14_0$ | 14 | **1608 [30]** | 311 |
| TG $16_0 16_0 16_0$ | 16 | 1926 [b] | 369 |
| TG $18_0 18_0 18_0$ | 18 | 2288 [b] | 436 |

[a] The $N_C$ is the number of the carbon atoms in the single side chain.    [b] Calculated with help of equation $C_{p,m}^o$(liq, 298.15 K) = 4.7021 $\times N_C^2$ + 21.0 $\times N_C$ + 387.4 with $R^2$ = 0.997, which was derived by approximation of experimental heat capacities given in this table in bold.   [c] Calculated with help of equation $-\Delta_l^g C_{p,m}^o$ = 16.4 + 0.1833 $\times C_{p,m}^o$(liq, 298.15 K) with $R^2$ = 0.979, which was derived by approximation of experimental $\Delta_l^g C_{p,m}^o$ -values for aliphatic long-chained esters [3].

As can be seen from Table 3, the data available are very limited, so it makes sense to approximate the available data as a function of chain length and use interpolation and extrapolation to estimate the heat capacities required. For homologous series, a linear correlation of the $C_{p,m}^o$(liq)-values with chain length is usually expected. For example, a good quality correlation was found for the linear aliphatic esters (see Table S6). To our surprise, the dependence of the heat capacity on the chain length for triglycerides is not linear and was approximated with the following polynomial:

$$C_{p,m}^o(\text{liq, 298.15 K}) = 4.7021 \times N_C^2 + 21.0 \times N_C + 387.4 \text{ (with } R^2 = 0.9972) \quad (4)$$

This correlation was used to estimate the missing heat capacities of triglycerides (see Table 3) and finally the heat capacity differences, $\Delta_l^g C_{p,m}^o$, for each triglyceride were calculated using Equation (3). The latter values and Equation (1) have enabled the uniform adjustment of our own and the literature data to the reference temperature $T$ = 298.15 K and these $\Delta_l^g H_m^o$(298.15 K) results are now available for comparison and evaluation (see Tables 1 and 2, column 5).

### 3.3. Evaluation of $\Delta_l^g H_m^o$(298.15 K)-Values of Triglycerides

A comparison of the $\Delta_l^g H_m^o$(298.15 K)-values for the relatively short chained triglycerides TG $2_0 2_0 2_0$, TG $3_0 3_0 3_0$, and for TG $4_0 4_0 4_0$ demonstrates generally good agreement for each molecule. Unfortunately, only single experimental values are available for TG $5_0 5_0 5_0$ and TG $7_0 7_0 7_0$, which makes these results questionable without further validation. For TG $6_0 6_0 6_0$ the range of available experimental vaporization enthalpies from 106 to 131 kJ mol$^{-1}$ makes it difficult to select a reliable value. The same ambiguity is for TG $8_0 8_0 8_0$, where the spread of the available experimental vaporization enthalpies ranges from 132 to 159 kJ mol$^{-1}$.

As can be seen from Table 2, the literature results of both TGA modifications (isothermal and non-isothermal) provide the higher and the lower values from this range of vaporisation enthalpies. In contrast to this, the results of the conventional static and tran-

spiration method, as well as from the ITGA method carried out in this work are definitely close to the lower level of the values collected for TG $8_0 8_0 8_0$. The same trend is also observed for TG $10_0 10_0 10_0$ where the static method, transpiration, QCM, and our ITGA show fairly similar results. At the same time, the literature modifications of the TGA provide significantly higher values (see Table 1). The wide spread of the literature TGA results can most likely be explained by the fact that this work [18–23] was published more than 20 years ago, when the development of this method for determining evaporation was still in its infancy. The process and the limits of the TGA method were not sufficiently known at the time. This statement is based on our extended investigation of the I-TGA method in relation to measurements with heavy volatile compounds [9]. In this work, we develop structure-property correlations to determine the general level of experimental enthalpies of vaporization. These correlations were very helpful in establishing consistency in the enthalpy of vaporization data for the entire TG set. For this reason, we have chosen to avoid averaging the 'experimental' and 'estimated' results. For example, for TG $7_0 7_0 7_0$, TG $12_0 12_0 12_0$ and TG $18_0 18_0 18_0$ the available experimental data did not agree with the general trend developed for the TG set. For this reason, averaging the "estimated" data was the only option to get the reasonable result for the particular TG.

The volatility of the triglycerides decreased dramatically with the lengthening of the alkyl chains, that is why all TGA results for TG $12_0 12_0 12_0$ provide unexpectedly high vaporisation enthalpies. The same conclusion applies for TG $14_0 10_0 10_0$ TG $16_0 10_0 10_0$ and TG $18_0 10_0 10_0$ (see Table 1). Therefore, for the latter series of the long-chained glycerides, only results from QCM and static methods (with the exception for TG $18_0 18_0 18_0$) could be considered as reliable.

For the triglycerides with the non-linear alkyl chains (branched or phenyl substituted) collected in Table 2, only single experimental values are available for each compound. Therefore, the quality of these results is questionable although the data were measured using conventional methods (transpiration, QCM, and static method).

### 3.4. Validation of Vaporisation Enthalpies

The significant disagreement among the available $\Delta_l^g H_m^o$(298.15 K)-datasets for most of the triglycerides compiled in Tables 1 and 2 has prompted the extended validation using structure–property correlations, e.g., with the chain length dependence or using the correlation between vaporisation enthalpy and retention indices, or boiling temperatures of triglycerides. Results of these validations are given below.

### 3.4.1. Structure–Property Correlations: Chain-Length Dependence

The linear correlation of $\Delta_l^g H_m^o$(298.15 K)-values with the number of carbon atoms in the alkyl chain within the homologue series of organic compounds is well-established phenomenon, e.g., for the series of aliphatic linear esters (see Table S6). We also correlated the $\Delta_l^g H_m^o$(298.15 K)-values for triglycerides (evaluated in Table 1) with the total number of carbon atoms, $n_c$, in the triglyceride. The following correlation was obtained (see Table 4):

$$\Delta_l^g H_m^o(298.15\ \text{K})/(\text{kJ·mol}^{-1}) = 50.7 + 3.18 \times n_c \text{ with } (R^2 = 0.9981) \quad (5)$$

The relatively high correlation coefficient $R^2$ was evidence of a good consistency of the experimental data evaluated in Table 1 and approximated by Equation (5).

As it apparent from Table 4 the differences between the experimental and "theoretical" vaporisation enthalpies are mostly below 3 kJ·mol$^{-1}$. The uncertainties of $\pm 3.0$ kJ·mol$^{-1}$ (0.95 level of confidence, k = 2) were assigned to the enthalpies of vaporisation, which were estimated from the correlation of $\Delta_l^g H_m^o$(298.15 K) with the number of C-atoms in the triglyceride. The "theoretical" results derived from Equation (5) are given in Table 1 labelled as $n_c$.

**Table 4.** Chain-length dependence of the experimental enthalpies of vaporisation, $\Delta_l^g H_m^o$ (298.15), for triglycerides with the linear saturated alkyl chains (in kJ mol$^{-1}$) [a].

| Compound | $n_c$ [a] | $\Delta_l^g H_m^o$(298.15 K)$_{exp}$ [b] | $\Delta_l^g H_m^o$(298.15 K)$_{calc}$ [c] | $\Delta$ [d] |
|---|---|---|---|---|
| TG $2_0 2_0 2_0$ | 9 | 81.5 ± 0.3 | 79.3 | 2.2 |
| TG $3_0 3_0 3_0$ | 12 | 90.4 ± 0.5 | 88.8 | 1.6 |
| TG $4_0 4_0 4_0$ | 15 | 97.5 ± 0.6 | 98.3 | −0.8 |
| TG $5_0 5_0 5_0$ | 18 | 104.8 ± 0.6 | 107.9 | −2.9 |
| TG $6_0 6_0 6_0$ | 21 | 114.9 ± 3.6 | 117.4 | −2.5 |
| TG $7_0 7_0 7_0$ | 24 | 127.0 ± 1.7 | 126.9 | 0.1 |
| TG $8_0 8_0 8_0$ | 27 | 134.1 ± 1.2 | 136.5 | −2.4 |
| TG $10_0 10_0 10_0$ | 33 | 160.1 ± 0.9 | 155.5 | 4.6 |
| TG $12_0 12_0 12_0$ | 39 | 174.9 ± 2.1 | 174.6 | 0.3 |
| TG $14_0 14_0 14_0$ | 45 | 194.0 ± 2.1 | 193.6 | 0.6 |
| TG $16_0 16_0 16_0$ | 51 | 211.6 ± 1.7 | 212.7 | −1.7 |
| TG $18_0 18_0 18_0$ | 57 | 232.1 ± 2.1 | 231.7 | 0.4 |

[a] Total number of carbon atoms in the triglyceride. [b] Values evaluated and recommended (given in bold) in Table 1. Uncertainty of the vaporisation enthalpy is expressed as the expanded uncertainty (0.95 level of confidence, k = 2). [c] Difference between column 3 and 4 in this table. [d] Calculated using Equation (5) with the assessed expanded uncertainty of ±3.0 kJ·mol$^{-1}$.

### 3.4.2. Structure–Property Correlations: Correlation with the Retention Indices $J_x$

The correlation of $\Delta_l^g H_m^o$(298.15 K) with the retention indices is also well-known tool to establish internal consistency within a set of structurally parent compounds, particularly the homologous series. The linear correlations are typical for different classes of organic compounds, e.g., alkyl-imidazoles [31], alkylbenzenes [32], and nitriles [33]. We have correlated the vaporisation enthalpies evaluated in Table 1 for the linear triglycerides and in Table 2 for the branched triglycerides with the data on Kovats indices, $J_x$, available from the literature [15,34]. The compilation of the data used for this correlation is given in Table 5.

**Table 5.** Correlation of vaporisation enthalpies, $\Delta_l^g H_m^o$ (298.15 K), of triglycerides with their Kovats indices ($J_x$).

| Compound | $n_c$ [a] | $J_x$ [b] | $\Delta_l^g H_m^o$(298.15 K)$_{exp}$ [c] kJ·mol$^{-1}$ | $\Delta_l^g H_m^o$(298.15 K)$_{calc}$ [d] kJ·mol$^{-1}$ | $\Delta$ [e] kJ·mol$^{-1}$ |
|---|---|---|---|---|---|
| glycerol triformate | 6 | 1112 | | *73.9* | |
| TG $2_0 2_0 2_0$ | 9 | 1299 | 81.5 ± 0.3 | 80.3 | 1.2 |
| TG $3_0 3_0 3_0$ | 12 | 1567 | 90.4 ± 0.5 | 89.5 | 0.9 |
| TG $4_0 4_0 4_0$ | 15 | 1816 | 97.5 ± 0.6 | 98.1 | −0.6 |
| glycerol tri(2-methylpropanoate) | 15 | 1692 | 93.6 ± 0.7 | 93.8 | −0.2 |
| TG $5_0 5_0 5_0$ | 18 | 2089 | 104.8 ± 0.6 | 107.5 | −2.7 |
| glycerol tri(3-methylbutanoate) | 18 | 1953 | 104.0 ± 1.3 | 102.8 | 1.2 |
| glycerol tri(2,2-dimethylpropanoate) | 18 | 1759 | | *96.1* | |
| TG $6_0 6_0 6_0$ | 21 | 2363 | 114.9 ± 3.6 | 116.9 | −2.0 |
| TG $7_0 7_0 7_0$ | 24 | 2664 | | *127.2* | |
| TG $8_0 8_0 8_0$ | 27 | 2958 | 134.1 ± 1.2 | 137.4 | −3.3 |
| TG $10_0 10_0 10_0$ | 33 | 3504 | 160.1 ± 0.9 | 156.1 | 4.0 |
| TG $12_0 12_0 12_0$ | 39 | 4050 | | *174.9* | |
| TG $14_0 14_0 14_0$ | 45 | 4604 | 196.4 ± 12.1 [16] | 194.0 | 2.4 |
| TG $16_0 16_0 16_0$ | 51 | 5158 | 210.2 ± 9.2 [Table S2] | 213.0 | −2.8 |
| TG $18_0 18_0 18_0$ | 57 | 5713 | | *232.1* | |

[a] Total number of carbon atoms in the triglyceride. [b] Kovats indices, $J_x$, on the non-polar column OV-101 [15,34]. [c] Experimental data evaluated in Tables 1 and 2. Uncertainty of the vaporisation enthalpy is expressed as the expanded uncertainty (0.95 level of confidence, k = 2). [d] Calculated using Equation (6) with the assessed expanded uncertainty of ±3.0 kJ·mol$^{-1}$. Values given in italic were used for comparison with the expeimenetal results in Table 1. [e] Difference between column 4 and 5 in this table.

The following linear correlation was found between the $\Delta_l^g H_m^o$(298.15 K)-values of triglycerides and the $J_x$-values:

$$\Delta_l^g H_m^o (298.15 \text{ K}) / (\text{kJ·mol}^{-1}) = 35.6 + 0.0344 \times J_x \text{ with } (R^2 = 0.9972) \qquad (6)$$

The high correlation coefficient $R^2$ of Equation (6) indicated the reliability of the vaporisation enthalpies evaluated in Tables 1 and 2. We used Equation (6) to predict vaporisation enthalpies of five triglycerides (given in Table 5 in italic), where retention indices were available, but the experimental data were of questionable quality. As is obvious from Table 5, the differences between the experimental and "theoretical" vaporisation enthalpies are mostly below 3 kJ·mol$^{-1}$. Therefore, the uncertainties of $\pm 3.0$ kJ·mol$^{-1}$ (0.95 level of confidence, k = 2) were assigned to the enthalpies of vaporisation, which are estimated from the correlation of $\Delta_l^g H_m^o$(298.15 K) with Kovats indices. The "theoretical" results derived from Equation (6) are given in Table 4 and labelled as $J_x$.

3.4.3. Structure–Property Correlations: Correlation with Normal Boiling Temperatures $T_b$

The correlation of $\Delta_l^g H_m^o$(298.15 K) with normal boiling temperatures, $T_b$, was additionally examined to validate vaporisation enthalpies of triglycerides evaluated in Tables 1 and 2. Such a correlation is usually expected to be linear, particularly within the homologous series. The triglycerides are generally thermally stable compounds; however, due to very high boiling points, data at standard pressure are limited. The normal boiling temperatures, $T_b$, of triglycerides have been compiled from the literature [35,36]. The data taken into correlation and results are given in Table 6.

**Table 6.** Correlation of vaporisation enthalpies, $\Delta_l^g H_m^o$ (298.15 K), of triglycerides with their normal boiling temperatures ($T_b$).

| Compound | $T_b$ [a] K | $\Delta_l^g H_m^o$(298.15 K)$_{exp}$ [b] kJ·mol$^{-1}$ | $\Delta_l^g H_m^o$(298.15 K)$_{calc}$ [c] kJ·mol$^{-1}$ | $\Delta$ [d] kJ·mol$^{-1}$ |
|---|---|---|---|---|
| TG $2_0 2_0 2_0$ | 533 [35] | $81.5 \pm 0.3$ | 79.3 | 2.2 |
| TG $3_0 3_0 3_0$ | 563 [35] | $90.4 \pm 0.5$ | 90.1 | 0.3 |
| TG $4_0 4_0 4_0$ | 583 [36] | $97.5 \pm 0.6$ | 97.3 | 0.2 |
| glycerol tri(3-methylbutanoate) | 606 [36] | $104.0 \pm 1.3$ | 105.5 | $-1.5$ |
| TG $6_0 6_0 6_0$ | 633 [36] | $114.9 \pm 3.6$ | 115.2 | $-0.3$ |
| TG $7_0 7_0 7_0$ | 665 [36] | | *126.7* | |
| TG $8_0 8_0 8_0$ | 691 [36] | $134.1 \pm 1.2$ | 136.0 | $-2.1$ |
| TG $16_0 16_0 16_0$ | 894 [36] | $210.2 \pm 9.2$ [Table S2] | 208.9 | 1.3 |

[a] Normal boiling temperature. [b] Experimental data evaluated in Tables 1 and 2. Uncertainty of the vaporisation enthalpy is expressed as the expanded uncertainty (0.95 level of confidence, k = 2). [c] Calculated using Equation (7) with the assessed expanded uncertainty of $\pm 3.0$ kJ·mol$^{-1}$. Value given in italic were used for comparison with the expeimenetal results in Table 1. [d] Difference between column 4 and 5 in this table.

It has been found that the $\Delta_l^g H_m^o$(298.15 K)-values of triglycerides are also linearly correlated with the $T_b$-values:

$$\Delta_l^g H_m^o (298.15 \text{ K}) / (\text{kJ·mol}^{-1}) = -112.1 + 0.3591 \times T_b \text{ with } (R^2 = 0.9989) \qquad (7)$$

The high correlation coefficient $R^2$ in Equation (7) supports the reliability of vaporisation enthalpies evaluated in Tables 1 and 2. Indeed, the vaporisation enthalpies derived from the correlation with the boiling temperatures are in a good agreement with the experiment. This good agreement can therefore be considered as an additional validation of the experimental data for the $\Delta_l^g H_m^o$(298.15 K) evaluated in this work (see Tables 1 and 2). The differences between the theoretical and experimental values are at the level of 2 kJ·mol$^{-1}$. However, considering a very limited set of experimental data included in the correlation, the uncertainties in the enthalpies of vaporisation estimated from the $\Delta_l^g H_m^o$(298.15 K)—$T_b$ correlation were evaluated to be $\pm 3.0$ kJ·mol$^{-1}$ (0.95 level of confidence, k = 2). We used Equation (7) to assess the vaporisation enthalpy of TG $7_0 7_0 7_0$ (given in Table 5 in italic),

where the experimental result seems to be inconsistent with other available data. The results of the $\Delta_l^g H_m^o$(298.15 K)—$T_b$ correlation are labelled as $T_b$ and given in Tables 1 and 2 for general comparison of the methods.

Finally, three independent structure–property correlations of vaporisation enthalpies with chain length, retention indices, and boiling temperatures have demonstrated sufficient internal consistency of the data analysis performed in Tables 1 and 2. These results are valuable for validating the level of enthalpies of vaporisation available from other methods, particularly for long-chain triglycerides, where the experimental data are in disarray. As can be seen in Tables 1 and 2, the results of the structure–property correlations labelled $n_c$, $J_x$ and $T_b$ agree, within the assigned uncertainties, with the evaluated experimental data (highlighted in bold). Therefore, these evaluated $\Delta_l^g H_m^o$(298.15 K)-values and given in bold can be recommended for thermochemical calculations.

### 3.5. Can the Group Additivity Method Predict Vaporisation Enthalpies of Triglycerides?

Group additivity (GA) methods are also a type of structure–property relationships [37,38]. The enthalpies of vaporisation of a set of molecules with reliable data are usually split up into the smallest possible groups, like "LEGO®" building blocks. Using the matrix calculations, each group obtains a well-defined numerical contribution. The prediction of the vaporisation enthalpy is then a construction of the molecule from the building blocks, collecting the energetics of a molecule from the appropriate number and type of bricks. In general, using this method for large molecules is impractical due to too many building blocks. To overcome this disadvantage, we develop a general approach to estimate vaporisation enthalpies based on a so-called "centerpiece" molecule [39,40]. The idea of the "centerpiece" approach is to start the prediction with a potentially large "core" molecule that can generally mimic the structure of the molecule of interest, but at the same time must have a reliable vaporisation enthalpy. The triglycerides are predestined for such an approach. The visualisation of the "centerpiece" approach for triglycerides is given in Figure 2.

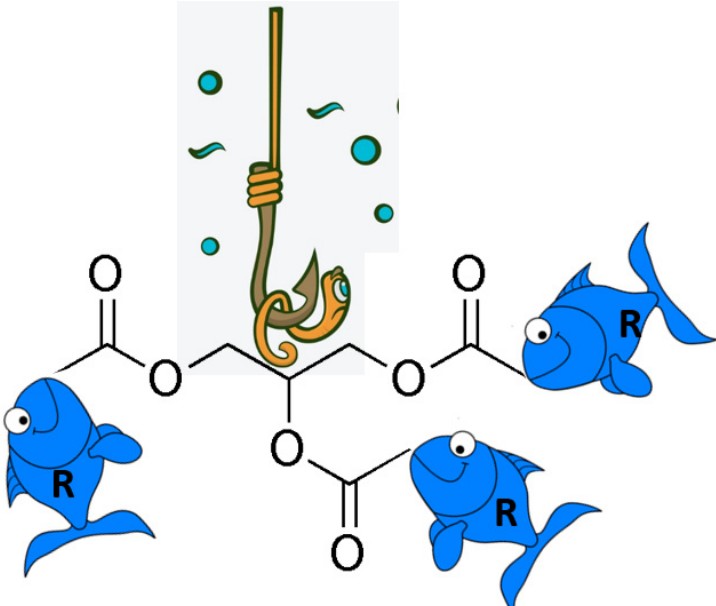

**Figure 2.** The visualisation of the "centerpiece" approach for triglycerides.

Indeed, the TG $2_02_02_0$ as the "centerpiece" model already bears the main energetic contributions to the vaporisation enthalpy specific for triglycerides. In order to obtain the "centerpiece" suitable for GA calculations we need only to cut three methyl groups (C-(C)(H)$_3$) from the TG $2_02_02_0$ (see Figure 3, left).

$$F_{TG} = \Delta_l^g H_m^o(298.15 \text{ K, glycerol triacetate}) - 3 \times C\text{-}(C)(H)_3$$

$$\Delta_l^g H_m^o(298.15 \text{ K, glycerol trihexanoate}) = F_{TG} + 3 \times C\text{-}(C)(H_2)(CO_2) + 9 \times C(C)_2(H)_2 + 3 \times C\text{-}(C)(H)_3 + 3 \times (C\text{-}CO)_{1\text{-}4} + 6 \times (C\text{-}C)_{1\text{-}4.}$$

**Figure 3.** Development of the "centerpiece" fragment $F_{TG}$ from glycerol triacetate (**left**) and a group-additivity calculation of $\Delta_l^g H_m^o$(298.15 K) for glycerol trihexanoate (TG $6_0 6_0 6_0$) as an example (**right**).

Such a bulk fragment (TriGlycerides or $F_{TG}$) and its energetic contribution are specific for triglycerides and hardly can be captured by any other method. This special feature of the "centerpiece" approach significantly increases the reliability of the property prediction for similarly shaped molecules, e.g., TG $6_0 6_0 6_0$ (see Figure 3, right), where substituents with the known contributions to the vaporisation enthalpy are simply attached to the **$F_{TG}$** as the "centerpiece". The group contribution values, which are specific for alkanes C-(C)(H$_3$), C-(C$_2$)(H$_2$), as well as the contribution C-(C)(H)$_2$(CO$_2$) specific for the methylene-group attached to the carbonyl-group are well established [14]. The contributions (C-C)$_{1\text{-}4}$ and (C-CO)$_{1\text{-}4}$ are additional correction terms for the 1–4 "gauche" interactions of carbon atoms along the alkyl chain. The details on these 1–4 C-C interactions are described elsewhere [38,41]. The schematic representation of the group-contributions involved in this study is shown in Table 7. The numerical values for these contributions were developed in our previous work [14] and are also given in Table 7.

**Table 7.** Group additivity contributions for calculation of the enthalpy of vaporisation, $\Delta_l^g H_m^o$ (298.15 K), for triglycerides with the saturated alkyl chains (in kJ mol$^{-1}$).

| | Increment [a] | $\Delta_l^g H_m^o$ [14] |
|---|---|---|
| | $F_{TG}$ | 62.5 |
| | C-(C)(H)$_2$(CO$_2$) | 3.2 |
| | C-(C)$_2$(H)(CO$_2$) | −1.5 |
| | C-(C)$_3$(CO$_2$) | −7.5 |
| | (C-CO)$_{1\text{-}4}$ | −1.0 |
| | C-(C)$_2$(H)$_2$ | 4.52 |
| | C-(C)(H)$_3$ | 6.33 |
| | (C-C)$_{1\text{-}4}$ | 1.80 |
| | C$_d$(H) | 3.8 |
| | Ph(CO$_2$) | 29.4 |

[a] For calculation of unsaturated alkyl-chains an additional increment C$_d$(H) = 3.8 kJ mol$^{-1}$ was developed from vaporisation enthalpies of olefines [38]. For calculation of phenyl substituted triglycerides an additional increment Ph(CO$_2$) = 29.4 kJ mol$^{-1}$ was developed from vaporization enthalpy of benzyl acetate [16].

Using the group additivity contributions given in Table 7, the predicted $\Delta_l^g H_m^o$(298.15 K)-values for triglycerides under study were calculated and the results are given in Tables 2, 8 and S7).

Even a quick look at results given in Table 8 can reveal that the "centerpiece" approach systematically overestimates the vaporisation enthalpies. It is noticeable that the overestimation is increasing with the growing chain-length of the triglycerides. It is obvious that the GA method has completely failed to predict the vaporisation enthalpies of

triglycerides. Is there an explanation for this phenomenon? The answer is discussed in the following section.

**Table 8.** Comparison of experimental and additive enthalpies of vaporisation, $\Delta_l^g H_m^o$ (298.15 K), for triglycerides with the linear saturated alkyl chains and for methyl alkanoates (in kJ mol$^{-1}$).

| | Triglycerides | | | Methyl Alkanoates | | |
|---|---|---|---|---|---|---|
| $N_c$ [a] | $\Delta_l^g H_m^o$(exp) [b] | $\Delta_l^g H_m^o$(add) [c] | $E_{disp}$(TG) [d] | $\Delta_l^g H_m^o$(exp) [e] | $\Delta_l^g H_m^o$(add) [f] | $E_{disp}$(MA) [g] |
| 3 | 90.4 | 91.1 | −0.7 | 36.0 | 35.8 | 0.2 |
| 4 | 97.5 | 101.6 | −4.1 | 39.6 | 40.3 | −0.7 |
| 5 | 105.0 | 115.9 | −10.9 | 43.6 | 45.1 | −1.5 |
| 6 | 114.9 | 130.2 | −15.3 | 48.5 | 49.8 | −1.3 |
| 7 | 127.0 | 144.4 | −17.4 | 53.4 | 54.6 | −1.2 |
| 8 | 134.1 | 158.7 | −24.6 | 57.2 | 59.4 | −2.2 |
| 10 | 160.1 | 187.3 | −27.2 | 66.5 | 69.0 | −2.5 |
| 12 | 174.9 | 215.8 | −40.9 | 75.5 | 78.5 | −3.0 |
| 14 | 194.0 | 244.4 | −50.4 | 85.2 | 88.1 | −2.9 |
| 16 | 211.6 | 273.0 | −61.4 | 93.6 | 97.6 | −4.0 |
| 18 | 232.1 | 301.5 | −69.4 | 103.7 | 107.2 | −3.5 |

[a] The number of C-atoms in a single alkyl chain in the triglyceride or in methyl alkanoates. [b] Evaluated values from Table 1. [c] Additive values calculated using increments in Table 7. [d] Difference between column 2 and 3, interpreted as amount of dispersion forces in triglyceride. [e] Experimental values evaluated in our previous study [3]. [f] Additive values calculated using increments in Table 7. [g] Difference between column 5 and 6, interpreted as amount of dispersion forces in methyl alkanoates (MA).

### 3.6. Non-Covalent Dispersion Interactions in Triglycerides

As a matter of fact, the GA methods are not only a suitable tool to predict molecular energetics, but also a tool to detect unusual energetic effects. When the experimental and additive results show significant discrepancies, it is best to look for specific interactions causing the deviation from additivity (assuming the experimental result is reliable). In the case of triglycerides, we validated the experimental vaporisation enthalpies with different structure–property correlations. In what follows, the profound deviation from additivity observed in Section 3.5 should only be caused by overlooked interactions specific to these long-chained molecules. Basically, the standard molar vaporisation enthalpy, $\Delta_l^g H_m^o$, is the portion of energy (enthalpy) required to transfer 1 mole of the liquid compound to a gaseous state. Thus, the vaporisation enthalpy can be taken as a measure of the overall attractive forces between the molecules in the liquid state. If these attractive interactions are responsible for significant interlinking of alkyl chains in the liquid phase, the energy required to take out the triglyceride with the interlocked chains from the liquid to the gas phase should be higher and the corresponding enthalpy of vaporisation greater in comparison to the additive value. Therefore, the relative decrease in the experimental vaporisation enthalpy can only be explained by assuming that the attractive forces are partially entrained into the gas phase. In this case, the attractive dispersion interactions between chains in the gas phase could be a plausible explanation for the deviation from additivity, since these specific non-covalent dispersion interactions are not considered in the GA parameterization. The existence of such dispersion-stabilized conformers in the gas phase has been theoretically supported by quantum chemical calculations [42]. Structural optimization of the long-chain triglycerides with MOPAC-PM7 showed that the most stable forms are folded conformers with three parallel chains that interact [43]. Similar to linear alkanes, folded configurations are favoured over extended star conformers [44,45]. Two possible structures of dispersion-stabilized conformers are shown in Figure 4, where the attraction of alkyl chains in the gas phase due to dispersion forces is evident.

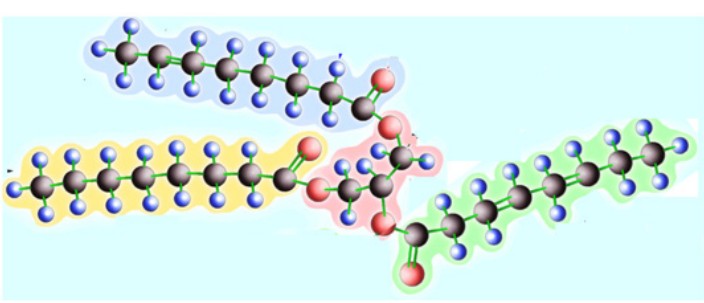
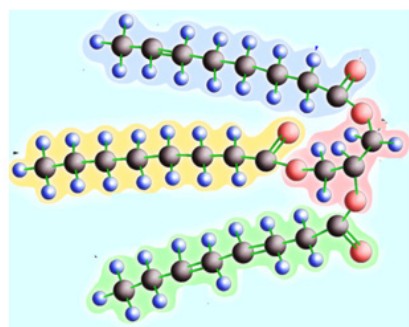

**Figure 4.** Examples of possible dispersion-stabilized conformations for TG $8_0 8_0 8_0$.

After qualitatively demonstrating the existence of dispersion forces, we used the results in Table 8 to quantitatively assess the strength of this interaction. To quantify the dispersion interactions in triglycerides, we assume that the differences between experimental enthalpies of vaporisation and additive values reflect the amount of non-additive forces (denoted as $E_{disp}$, see Table 8 and Figure 5).

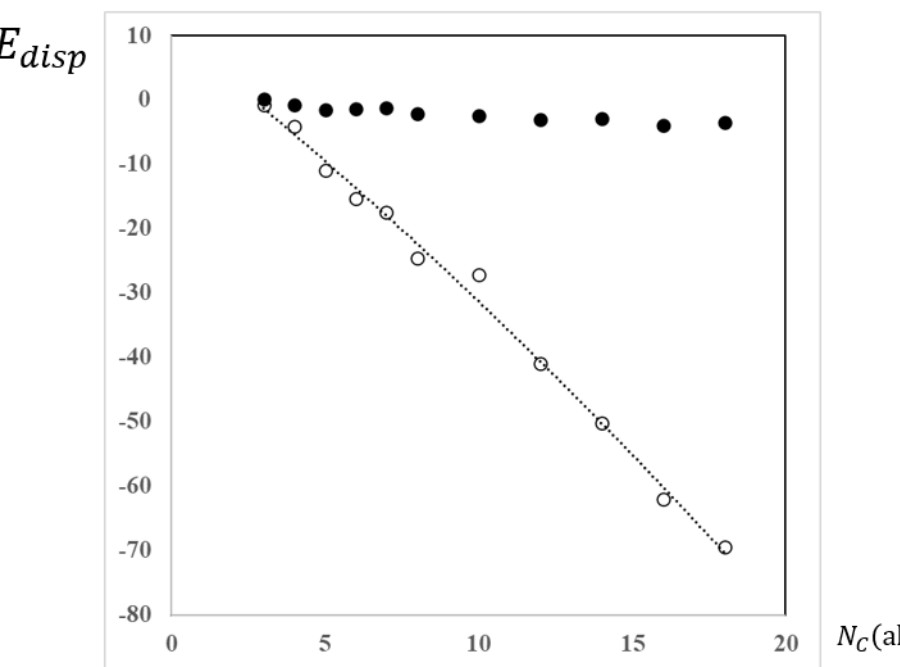

**Figure 5.** Comparison of amount of dispersion interactions (in kJ mol$^{-1}$) in triglycerides (○) and in methyl alkanoates (●).

It is quite obvious that the differences, $E_{disp}$, do not represent the total energy of the dispersion forces between alkyl chains in triglycerides. Nonetheless, $E_{disp}$-values can be considered as an energy differences originating from dispersion forces between the three arms of the triglyceride, and the vaporisation enthalpies provide a reliable measure of dispersion forces in triglycerides. Therefore, the dramatic increase (from $-0.7$ kJ mol$^{-1}$ for TG $3_0 3_0 3_0$ to $-69.4$ kJ mol$^{-1}$ in TG $18_0 18_0 18_0$) in dispersion interactions with growing chains length can now be conceptually explained and understood. Indeed, in a homologous series of methyl alkanoates, although the enthalpies of vaporisation logically increase with growing chain length (see Table 8 and Figure 5), the differences between the experimental and additive values that account for dispersion interactions hardly increase with increasing chain length. Thus, the dispersion interactions appear to be negligible for methyl alkanoates but not in triglycerides.

How does alkyl chain branching affect dispersion interactions between alkyl chains in the triglycerides? To answer this question, we calculated the additive enthalpies of vaporisation, $\Delta_l^g H_m^o$(298.15 K), for all entries in Table 2. A comparison of the recommended (given in bold) experimental and additive enthalpies of vaporisation, $\Delta_l^g H_m^o$(298.15 K), reveals that the values for glycerol tri(2-methylpropanoate) and glycerol tri(3-methylbutanoate) are very close (within their uncertainties, hence the dispersion forces are small, since the branching of the chains precludes close approximation of chains. The difference between additive and experimental values of $-6.8$ kJ mol$^{-1}$ for glycerol tri(2,2-dimethylpropanoate) indicates noticeable stabilization despite the steric interactions of bulky substituents. However, such an interesting phenomenon has already been considered specific, as shown for tert-butyl-substituted alkanes [2]. For the glycerol tribenzoate the stabilization of $-9.3$ kJ mol$^{-1}$ could be explained by the $\pi$–$\pi$ attractive interaction of the benzene rings attached to the TG moiety. The profound stabilization of $-44$ kJ mol$^{-1}$ observed for glycerol trierucate (TG $22_1,22_1,22_1$) appears due to dispersive interactions of the long alkyl chains. However, this stabilization is less intensive than expected from the E$_{disp}$ trend obvious from Table 8. It appears that the double bonds present in the alkyl chains screw them and reduce the spatial possibilities for the attractive dispersion forces.

To draw a practical conclusion from this study, we failed to develop the group contribution method for predicting the vaporisation enthalpies of triglycerides because the non-covalent dispersion interactions are unique to each triglyceride and increase with increasing chain length. Nonetheless, the critical evaluation and validation of the vaporisation thermodynamics of triglycerides has enabled a qualitative and quantitative understanding of the reasons for these dispersion forces.

**Supplementary Materials:** The following supporting information can be downloaded at: https://www.mdpi.com/article/10.3390/thermo2030018/s1, Figure S1: Schematic explanation of the TG abbreviations; Table S1: Results of transpiration method for triglycerides measured in this work: absolute vapour pressures $p$, standard molar vaporisation enthalpies and standard molar vaporisation entropies; Figure S2: The scheme of the QCM experimental setup from; Table S2: The experimental mass loss rates and vapour pressures determined for TG$10_010_010_0$ and TG$16_016_016_0$ with QCM technique; Table S3: The experimental mass loss rates and vapour pressure determined for TG$8_08_08_0$ and TG$10_010_010_0$ with the I-TGA technique; Table S4: Vapour pressures $p$, standard ($p^o$ = 0.1 MPa) molar vaporisation enthalpies, $\Delta_l^g H_m^o$, and standard ($p^o$ = 0.1 MPa) molar vaporisation entropies, $\Delta_l^g S_m^o$ obtained by the approximation of data collected from SciFinder; Table S5: Compilation of data on molar heat capacities $C_{p,m}^o$(liq) and heat capacity differences $\Delta_l^g C_{p,m}^o$ for the linear aliphatic esters at $T$ = 298.15 K (in J.K$^{-1}$.mol$^{-1}$); Table S6: The chain length dependence of the molar heat capacities $C_{p,m}^o$(liq) for the linear aliphatic esters at $T$ = 298.15 K (in J.K$^{-1}$.mol$^{-1}$); Table S7: Comparison of experimental and additive enthalpies of vaporisation, $\Delta_l^g H_m^o$(298.15 K), for triglycerides with the linear saturated alkyl chains (in kJ.mol$^{-1}$) [3,4,6,8,9,46–51].

**Author Contributions:** Conceptualization, S.P.V. and R.N.N.; methodology, S.P.V. and R.N.N.; validation, R.N.N.; formal analysis, R.N.N.; writing—original draft preparation, S.P.V. and R.N.N.; writing—review and editing, S.P.V. and R.N.N.; funding acquisition, S.P.V. All authors have read and agreed to the published version of the manuscript.

**Funding:** EU Project "Metrology of biofuels", German Science Foundation (DFG) in the frame of SPP 1807 "Control of London Dispersion Interactions in Molecular Chemistry" (grant VE 265-9/2), and Kazan Federal University Strategic Academic Leadership Program ("PRIORITY-2030").

**Data Availability Statement:** Data supporting reported results can be found in the supplementary materials to this paper.

**Acknowledgments:** We gratefully acknowledge the contributions of V.N. Emel´yanenko and D.H. Zaitsau for assistance in the thermochemical experiments.

**Conflicts of Interest:** The authors declare no conflict of interest.

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
