# Peer review of "Non-Covalent Interactions in Triglycerides: Vaporisation Thermodynamics for Quantification of Dispersion Forces"

_2673-7264, doi:10.3390/thermo2030018_

Round 1

Reviewer 1 Report

Major comments:

·        Aim of the publication is not clear, its either (i) an accurate compilation of experimental data on vaporization enthalpy of triglycerides from different sources, or (ii) prediction of vaporization enthalpy of triglycerides using GA method (in last case, concept of training and test sets should be more elaborated).

·        Authors need to provide more literature evidence about gas-phase dispersion in TG (in turn, dispersion of solid in liquid is well-known in TG). One component phase diagram would be of help. Statement of non-covalent dispersion interactions in TG in gas-phase would require more supporting data (e.g. additional ab-initio calculations leaded by correlation between observed error of GA predictions and energy for various gas-phase TG conformations).

Minor comments:

·        More reference in introduction is required

·        Figure 1 (or its cation) seems to be mixed up with Figure S1

·        Experimental data obtained for 4 linear TG in the work are missing in Table 1 (what the main purpose of this measurements?)

·        Table 1: “method” can’t stay as “n/a”, it does not correspond to description of carefully collected data

·        Table 1: why data from Eq.5-7 are not always used for “mean” enthalpy calculations?

·        Table1: some experimental points (in brackets) were skipped, corresponding explanation is required in the main text

·        Table 1: reference should be on published/deposited data not on supporting information

·        Table 2: compounds with “this work” label are missing in “Materials and methods”

·        It would be very helpful to support correlations Eq.5-7 with corresponding plots

·        Its better to use “predicted” instead of “theoretical” enthalpy for GA data

·        Table 7: side chains representation is confusing, use 2D sketch instead (and may be merge table 7 with Figure 4)

·        Add full list of chemical classes which were used to derive side chain corrections (table 7), it would be helpful to   add them in Table 8 as well

·        Table 8: for now I would recommend to change “E_disp” to “error of prediction”

Reviewer 2 Report

Line 11: English, is reliable should read is a reliable

Line 18: there is no closing bracket: relationships (chain-length

Line 22: the comma should be taken out here: turned out, that the family

Same (delete comma) in

Line 371 ‘It makes oneself conspicuous, that the overestimation is increasing’

Line 35: As a physical chemist I doubt that ‘Dispersion forces are considered to be the least studied phenomena’. They are much more difficult to handle so to speak, and therefore less is known about them, in particularly quantitatively as the author’s state correctly though.

Line 36: ‘The dispersion forces are usually related to the attractive part of the van der Waals potential [1].’ In my opinion a more general, older, e.g. textbook reference would suit better, as it not the authors who found this.This is what I meant when marking (above) Did you detect inappropriate self-citations by authors?

Fig.1: the caption is missing, and figure 1 both left hand and right hand part in fact show the same, I suggest to delete the left hand part

After Fig 1 there is a sentence starting with Figure s1, and the subsequent sentence does start strange, as if initial words are missing, and it does not read like the caption of Figure S1. Please correct.

Table 1: for chemists to read this table more easily it would be useful to name the triglycerides by their name, e.g. triacetin, glycerol triacetate. The abbreviations like TG 202020 used by the authors is referenced as [1]               Brinkmann, J.; Luebbert, C.; Zaitsau, D. H.; Verevkin, S. P.; Sadowski, G., Thermodynamic Modeling of Triglycerides using PC-SAFT. J. Chem. Eng. Data 2019, 64 (4), 1446-1453 is as far as I know, and I did an additional search, not a known and accepted naming for these compounds, In Table 2 the authors did use the common chemical namings, but for one and only one also added the TG nomenclature. Please make consistent. Same comment for Table 3, Table 5.

Table 1 and Table 2: I do not think the authors well-indicated why they put some values in bold. I think it is obvious, but the authors should comment, also why (Table 1) some values were not incorporated in  the averages and others are not.

Line 147: ‘Perhaps, the reason for the deviation of the empirical coefficients from those of the original values is that not too many long-chain species were included in the evaluation of Chickos and Acree [4; 28].’ As I think Chicos is all a well-known scientist in  the field, it would make sense to find out what really is the reason for the difference, in particular because the authors wrote ‘Both empirical coefficients are significantly lower than the original values from Chickos and Acree [4; 28]’ First of all it could be explicitly mentioned how and which long-chain species are currently considered (compared to Chicos and Acree), but in particularly what would happen with the coefficient if Verevkin and Namigranov would omit these longer species in their list and redetermine the coefficients, would they still differ from Chicos’ values or not?

Line 292: is it correct to referee to Eq.(6) ‘The high correlation coefficient R2 in Eqs. 6 ‘ as this sentence follows Eq.(7)

Line 317: English: like a “LEGO bricks” should read like LEGO bricks’. However, reading further the wording bricks should be avoided, this is not a common terminology in GC methodology of molecules. Small building blocks like CH2, CH3, or momers, or alike, could be mentioned. Or a small graphical illustration. Either one is familiar with the concept and the explanation is superfluous, or one is not and a graphical ‘ explanation’  says more than wordings.

Table 8, in the caption it reads methyl alkanoates but in the table esters, please make consistent.

In Table 8, for the esters there is also a column Edisp. Following the authors discussion on the triglyceries at this point, the differences exp-theor should not be named Edisp for the esters because there is not such a term, the differences are the ‘ usual’  small differences between exp and good model values. In fact for the sake of discussion it might also be better to name them a difference for the triglycerides, and in  the text argue why these are dispersion energy contributions. Moreover, the authors themselves state ‘It is quite obvious that the differences, Edisp, do not represent the total energy of the dispersion forces between alkyl chains in triglycerides.’ To me this makes the terminology Edisp somewhat misleading. So in fact what is meant is an energy differences originating from dispersion forces between the three arms of the triglyceride.

Figure 6 should be made quadratic or alike, now the x-axis is extremely extended and the y-axis compressed. Now it looks the more than all values for triglycerides are on the indicated curve, but there are still clear differences.

Coming back to the discussion around Table 8, the statement ‘the dramatic increase (from -0.7 kJ mol-1 for TG 303030 to -69.4 kJ mol -1 in TG 180180180) in dispersion interactions with growing chains length is now conceptually explained and understood.’ It seems a very logical explanation, also in the absence of another one, but I think it is not unfair to say it remains to be proven that it is the dispersion energies that are fully responsible. In that sense in would be interesting to see whether a similar, though quantitatively smaller, effect is present in ‘two-arm’  systems such as, e.g. isopalmitic acid (pentadecane-7-carboxylic acid) or the cyclic alkanes. For the linear alkanes I think it is well-known and understood that it is the long-range interactions are the cause of the behaviour of the Tb versus chain length: it is clearly not linear. However, for the triglycerides in this study the relation between Tb and chain length is pretty linear (up to C16), unlike the linear alkanes. To me this all suggests it is not yet that clear what accounts for the differences between exp and theor for the Hvap in the present study.

HOWEVER, in summary I do not want to suggest a lot of additional work should be done, but my suggestion is (unless the authors have clear quantitative arguments why they are right) clear make it less conclusive, but a clear suggestion which needs further clarification. A problem then is the clear claim in title and abstract that ‘vaporisation thermodynamics for quantification of dispersion forces’. If true, it is only part of the dispersion forces, but which part?

I think (at least to me) the paragraph starting at line 420 is a little obscure. How are these values calculated? This is not well explained. What is different from the numbers shown in Table 8 and Figure 6?

Round 2

Reviewer 2 Report

I thank the authors for the new version, it rally looks very good and satisfactory.

A minor thing: still not so happy with what was my

Point 10: Table 1 and Table 2: I do not think the authors well-indicated why they put some values in bold. I think it is obvious, but the authors should comment, also why (Table 1) some values were not incorporated in the averages and others are not

I fully accept the reply the authors wrote, but I do not see it well reflected in the caption of the figures in the manuscript's revised version